# COVID-19 as Another Trigger for HBV Reactivation: Clinical Case and Review of Literature

**DOI:** 10.3390/pathogens11070816

**Published:** 2022-07-21

**Authors:** Caterina Sagnelli, Laura Montella, Pierantonio Grimaldi, Mariantonietta Pisaturo, Loredana Alessio, Stefania De Pascalis, Evangelista Sagnelli, Nicola Coppola

**Affiliations:** Department of Mental Health and Public Medicine, Section of Infectious Diseases, University of Campania “Luigi Vanvitelli”, 80134 Naples, Italy; lauramontella1@gmail.com (L.M.); peogrimaldi@me.com (P.G.); mariantonietta.pisaturo@unicampania.it (M.P.); loredana.alessio@gimail.com (L.A.); strefaniadepascalis77@tiscali.it (S.D.P.); evangelista.sagnelli@unicampania.it (E.S.); nicola.coppola@unicampania.it (N.C.)

**Keywords:** COVID-19, SARS-CoV-2, Hepatitis B virus, reactivation, immunosuppression therapy, prevention

## Abstract

Universal hepatitis B virus (HBV) vaccination has been applied for years in most countries, but HBV infection remains an unresolved public health problem worldwide, with over one-third of the world’s population infected during their lifetime and approximately 248 million hepatitis B surface antigen (HBsAg) chronic carriers. HBV infection may reactivate with symptomatic and sometimes life-threatening clinical manifestations due to a reduction in the immune response of various origins, due to chemotherapy or immunosuppressive therapy, treatments increasingly practiced worldwide. SARS-CoV-2 and its COVID-19 associated disease have introduced new chances for HBV reactivation due to the use of dexamethasone and tocilizumab to counteract the cytokine storm. This could and should be prevented by accurate screening of HBV serologic markers and adequate pharmacologic prophylaxis. This article describes the case of a patient with COVID-19 who developed HBV reactivation and died of liver failure and analyzes published data on this setting to provide useful information to physicians who manage these patients during the SARS-CoV-2 pandemic.

## 1. Introduction

Patients who become infected with hepatitis B virus (HBV) infection develop a symptomatic or asymptomatic acute hepatitis that heals in most patients and evolves to chronicity in a minority of them at a frequency of 10–15% in children and 2–5% in adults.

Whatever the outcome, an indelible trace of HBV infection remains in the nucleus of infected hepatocytes, the covalently closed circular DNA (cccDNA) acting as a template for HBV replication and resistant to all available treatments. In fact, the high genetic barrier nucleo(t)ide analogues entecavir (ETV), tenofovir disoproxil fumarate (TDF), and tenofovir alafenamide fumarate (TAF) suppress HBV replication but fail to eradicate viral cccDNA which, once antiviral treatment is discontinued, works again as a template for a new HBV production. This may also occur in the case of a reduction in the immune response of various origins, including the high dose and/or long-term administration of corticosteroids [1,2,3,4,5,6,7,8,9]. Under these conditions, HBV reactivation may occur both in hepatitis B surface antigen (HBsAg)-positive subjects and in the HBsAg-negative/hepatitis B core antibody (HBc-Ab)-positive; HBV reactivation, however, is less frequent in HBsAg negative individuals [10,11,12,13]. COVID-19 guidelines suggest using dexamethasone at the dose of 6 mg daily for at least one week. In clinical practice, since respiratory failure lasts longer than 10 days, the time span in which corticosteroid use is safe is commonly exceeded [1,2,3,4,5,6,7,8].

Following HBV reactivation, the cytotoxic T-cells pre-sensitized to HBV proteins recognize the HBV core antigen (HBcAg) newly expressed on the surface of the hepatocytes and cause necro-inflammation, which is frequently symptomatic and seldom life threatening. Among the immunosuppressive conditions, COVID-19 offers a further possibility of HBV reactivation due to the dexamethasone and tocilizumab (TCZ) treatment administered to counteract the cytokine storm.

## 2. HBV and SARS-CoV-2 Clinical Interaction

Abnormalities in liver function have been observed in 14.8–53.0% of patients with COVID-19 [14,15,16]. Different pathogenetic mechanisms have been hypothesized as a possible cause of liver damage in this setting: the direct action of SARS-CoV-2 on liver cells, the cytokine storm, a hypoxic-ischemic liver injury, a drug-induced liver injury and the reactivation of a pre-existing liver disease [17,18,19,20,21,22].

It is still unclear whether COVID-19 can make chronic HBV hepatitis more severe, or whether people with pre-existing HBV infection are at greater risk of becoming infected with SARS-CoV-2. In a retrospective study performed by Zou et al. [23] on 105 patients with COVID-19 and HBV infection in 14 patients, their pre-existing HBV infection with liver injury worsened. Ten of them recovered in 6-21 days and four rapidly progressed to an acute-on-chronic liver failure. In this study, severe COVID-19 was more frequent in patients with liver injury who, in addition, more frequently developed serious complications, such as acute cardiac injury and shock.

Different results were obtained by Liu et al. [24] who studied 50 patients with COVID-19 and pre-existing HBV infection in comparison with 56 patients with COVID-19 alone and found similar percentages of moderate, severe, and critical patients in both groups and no difference in the outcome of COVID-19. Similarly, Chen et al. [25] compared 20 patients with COVID-19 and pre-existing HBV infection with 306 COVID-19 patients lacking HBV and found no differences in liver function and in the length of hospital stay between the two groups.

In summary, the interactions between pre-existing HBV infection and COVID-19 are not clear since a few conflicting studies have been published so far. Instead, it is important to understand if and to what extent the use of corticosteroids and immunosuppressants to control the cytokine storm in COVID-19 can lead to a reactivation of HBV infection, overt or occult.

In our clinical practice, we came across a dramatic HBV reactivation in an HBsAg-positive, HBV DNA negative patient with COVID-19 we preferred to include in this article the description of some clinical cases of HBV reactivation of different clinical severity, in the awareness that the study of single cases represents an educational model and complementary to a systematic education.

## 3. HBV Reactivation in an HBsAg-positive/HBV-DNA-negative Patient with COVID-19 Pneumonia

A 56-year-old man was hospitalized for COVID-19 pneumonia complicated by a pulmonary micro embolism at a COVID-19 center in Naples on day 0. The patient had been chronically infected with HBV since 1986 but had normal serum aminotransferases. The patient had never been treated with nucleos(t)ide analogues, and was without a history of alcohol use and concomitant causes for liver injury.

At baseline the patient presented as HBsAg positive, hepatitis B surface antibody (HBsAb) negative, hepatitis B core antibody (HBcAb) positive, hepatitis C virus antibody (HCV-Ab) negative, hepatitis B e antigen (HBeAg) negative, hepatitis B e antibody (HBeAb) positive, hepatitis Delta virus antibody (HDV-Ab) negative and human immunodeficiency virus (HIV) antibody negative, total bilirubin 1.2, alanine aminotransferase (ALT)/aspartate transaminase (AST) within normal range, creatinine 1.1 mg/dL, Na 135 mmol/L, K 3.4 mmol/L, platelets (PLT) 75,000 /uL, and an international normalized ratio (INR) of 1.1.

During hospitalization, he received dexamethasone 6 mg daily, low-molecular-weight heparin, and O2-therapy, with a clinical remission and SARS-CoV-2 clearance in 12 days, and no anti-SARS-CoV-2 antivirals were given.

The patient was discharged on day 14. At home, dexamethasone at the dose of 6 mg was administered for 12 days, then 4 mg for seven days, 2 mg for another seven days, 1 mg for another seven days, and stopped at day 26. The dosage was not administered pro kg as usually happens in acute respiratory distress syndrome (ARDS) (Figure 1).

Seven days after the complete discontinuation of corticosteroids, the patient developed acute autoimmune thrombocytopenia (idiopathic thrombocytopenic purpura) with PLT < 14,000 /uL). The patient underwent a blood smear to exclude other causes. He was then treated in a day hospital regimen, in another hospital, with methylprednisolone (1 mg/kg daily i.v. for one week) and a single i.v. administration of human immunoglobulin (0.8 g/kg), and subsequently descaled methylprednisone at home with the scheme of 1/2 dose for 3 days and 1/4 dose for 4 days (day 48).

The patient became icteric on day 55 and was admitted to another hospital where an abdominal CT scan examination showed an enlarged liver with no focal lesion, an ectasic portal vein thrombosed at the spleno-porto-mesenteric confluence with extension to the origin of the superior mesenteric and splenic veins, venous-perigastric collateral circulation and conspicuous abdominal-pelvic ascites. In the hypothesis of HBV reactivation, an empirical therapy with ETV at the dose of 1 mg/day was started on the same day (Figure 1).

On day 62, the patient was hospitalized in our liver unit in serious clinical condition, and was still receiving ETV on admission. At entry, laboratory tests showed a total bilirubin of 29.69 mg/d, ALT, and AST 6.3- and 9.7-times the highest value of normal, respectively, albumin 2.8 g/dL, INR 3.08, prothrombin time (PT) 34.3′’, activated partial thromboplastin time (aPPT) 52.8′’, d-dimer 3819 ng/mL, and fibrinogen 121 mg/dL.

Serological tests indicated that the patient was HBsAg-positive, HBsAb-negative, HBcAb-positive, HCV-Ab negative, HBeAg-negative, HBeAb-positive, HDV-Ab negative, and anti-HIV antibody negative.

HBV DNA declined from the baseline 6.1 × 10^3^ IU/mL to 499 IU/mL after seven days of ETV therapy, on day 62 and was not detected after 15 days of therapy (day 70) (Figure 1).

Supportive therapy for hepatic decompensation and monitoring of hematological, biochemical, and coagulation parameters in blood were also performed, but the clinical condition and laboratory parameters progressively worsened, hospital-acquired infections contraindicating liver transplantation occurred, and the patient died on day 108. The patient continued ETV until the time of death.

Supportive therapy for hepatic decompensation and monitoring of blood chemistry.

It is likely that COVID-19 and /or treatments used to counteract the cytokine storm may have played a role both in the onset of acute autoimmune thrombocytopenia, on the development of diffuse vascular damage, and on the development of hepatic injury. It is most likely that dexamethasone and other corticosteroids changed a host/HBV balance that was stable for decades and the reactivation of HBV is proof of this effect.

### Statement of Ethics

All procedures performed were in accordance with the international guidelines, with the Helsinki Declaration of 1975, revised in 1983, and the rules of the Italian laws of privacy, and with the local Ethics Committees named “Comitato Etico Universita’ Degli Studi Della Campania“Luigi Vanvitelli”—Azienda Ospedaliera Universitaria “Luigi Vanvitelli”—Azienda Ospedaliera Rilievo Nazionale “Ospedali Dei Colli””, Naples, Italy (n°10877/2020, 11 May 2020). On 4 August 2021, the patient signed an anonymous informed consent for the use of their data for anonymous clinical investigations and scientific publications. At the baseline visit, the patient provided informed consent for the surgical procedure.

## 4. Clinical Data from the Literature

A summary of published data on HBV reactivation in COVID-19 patients is shown in Table 1.

Wu et al. [26] evaluated a 45-year-old male with chronic HBV infection for over 20 years, initially treated with adefovir dipivoxil and ETV and subsequently with ETV alone [26].

This patient was hospitalized for COVID-19 and presented as HBsAg-positive, HBeAg-, HBeAb- and HBV DNA-negative, with ALT 1.2 times above the upper normal value (n.v.) and AST within the normal range. He received a six-day course of methylprednisolone followed by a moderate hepatic exacerbation with moderate increase in ALT, AST values and HBV DNA becoming detectable, albeit at low levels (1.11×10^2^ IU/mL). TDF was added to ETV therapy, the flare abated, and the patient recovered from COVID-19. The clinical course of the illness does not allow discrimination between COVID-19 and steroid administration as a possible cause of HBV reactivation [26].

Aldalheei et al. [27] described a 35-year-old man hospitalized for unconsciousness arising after some episodes of vomiting and having a Glasgow Coma Scale of 7/15, and was also icteric and positive for SARS-CoV-2 RNA on nasopharyngeal swabs and transferred to the Intensive Care Unit. Serum biochemistry showed AST 4933 IU/L (n.v. < 50), ALT 4758 IU/L (n.v. < 40), total bilirubin 10.75 mg/dL, conjugated bilirubin 8.75 mgdl, and INR >10. Revise to: The patient was HBsAg-, HBcAb IgM- and HBeAb-positive, HBeAg was negative and HBV DNA 2490 IU/mL. Brain CT scan and abdominal Doppler ultrasound did not show pathologic findings, and a chest X-ray was normal. Hepatitis A Virus, HCV, Hepatitis E Virus, autoimmune hepatitis, and Wilson’s disease were excluded. The patient was treated with lactulose through a nasogastric tube, ETV 1 mg daily, vitamin K 10 mg daily, and thiamin intravenously (100 mg daily). In 16 days, liver function improved, the SARS-CoV-2 PCR became negative, and the patient regained full consciousness.

The high HBV DNA level in an HBeAg-negative/antiHBe-positive patient was interpreted by the authors as indicative of HBV reactivation and vomiting episodes as an atypical initial symptom of COVID-19. They then speculated that SARS-CoV-2 played a role in HBV reactivation. We believe that in the absence of data on HBV serological markers prior to the reported events, the diagnosis of severe acute HBV infection concomitant with COVID-19 cannot be ruled out.

Rodrìguez-Tajes et al. [28] investigated 600 COVID-19 patients, of which data on HBV infection was available in 484, and of whom 69 (14%) screened HBsAg-negative, anti-HBc-positive, and HBV DNA undetectable. Of these 69, 61 could have been followed up for possible HBV reactivation. Anti-HBs > 10 UI/mL was detected in 44 (72%) of 61 patients. The investigated patients underwent immunosuppressive therapy with the IL-6 receptor antagonist TCZ, the IL-1 receptor antagonist anakinra, the Janus kinase inhibitor baricitinib and/or corticosteroids. Prophylaxis against HBV reactivation was strongly recommended for these 61 patients, but 38 received ETV 0.5 mg daily for one month and 23 remained untreated. In detail, Anti-HBs >10 UI/mL was detected in detail in 27 (71%) in patients in ETV prophylaxis, and in 14 (74%) in non ETV prophylaxis group.

As an unequivocal sign of HBV reactivation, HBV-DNA became detectable in serum in two (8.7%) of the 23 patients in the non-prophylaxis group and in none of the 38 who underwent ETV prophylaxis. The authors concluded that there was a low risk of HBV reactivation in HBsAg-negative/HBcAb-positive, HBV DNA negative patients treated for COVID-19, as it was preventable by adequate prophylaxis with nucleo(t)side analogues [28].

## 5. Prevention of HBV Reactivation in COVID-19 Patients

Patients with COVID-19 receive immunosuppressive drugs to control the cytokine response and reduce the immuno-mediated organ damage. The most used drugs in this setting are dexamethasone, at the dosage of 6 mg daily, and TCZ at the daily dosage of 8 mg/kg (maximum daily dosage 800 mg) in one administration soon after the start of respiratory support [29,30,31,32,33,34,35].

The existing data on the risk of HBV reactivation during COVID-19 are scarce, but those from other sectors (onco-hematological, rheumatological, gastroenterological) [36,37,38,39,40,41,42,43,44,45,46,47,48,49,50,51] strongly suggest that all patients, including those SARS-CoV-2 RNA-positive, should be screened for markers of HBV infection before starting corticosteroid and / or immunosuppressive treatments. Of course, duration and dosing for steroid treatment should be taken into account. HBsAg-positive patients should start prophylaxis with high genetic barrier antivirals (ETV, TDF, TAF) as soon as possible and continue for at least 12 months after stopping these drugs. Hepatic function and serum HBV DNA levels should be monitored at an interval of three months during drug prophylaxis and for another 12 months after its discontinuation. It is worthy of note that in one case HBV reactivation occurred after this period [52].

HBsAg-negative/anti-HBc-positive patients with COVID-19 are at low risk of HBV reactivation during immunosuppression since this treatment does not include anti-CD20 agents [8,14,15,20,23,29,32,33,34,35,36]. In this low-risk setting, pre-emptive therapy may be preferred to prophylaxis, with the timely start of nucleo(t)side analogues in the event that HBsAg becomes positive and/or HBV DNA detectable in serum. Such events should be identified early on by monitoring serum HBsAg and HBV DNA at one-to-three-month intervals.

The EASL guidelines recommend prophylaxis for HBsAg-negative/anti-HBc-positive patients only in cases of prolonged immunosuppression, limited compliance of patients to a long-term follow-up, or administration of new biological drugs [53].

The reported data highlight the need for prospective studies to improve our knowledge on the effects of corticosteroids and immunosuppressive drugs on HBV reactivation in patients with COVID-19. It is also of marked interest to gather information on how COVID-19 centers are addressing the problem of HBV reactivation and, if necessary, to provide them with the elements for the correct management of this problem.

## 6. Conclusions

Data on the risk of HBV reactivation during COVID-19 are scarce, but those from other sectors (onco-haematological, rheumatological, gastroenterological) [36,37,38,39,40,41,42,43,44,45,46,47,48,49,50,51] strongly suggest that all patients, including SARS-CoV-2 RNA-positive, should be screened for markers of HBV infection before starting corticosteroid and / or immunosuppressive treatments. HBsAg-positive patients should earlier start prophylaxis anti-HBV reactivation with high genetic barrier antivirals and continue for at least 12 months after stopping these therapies.

The use of TCZ seems less dangerous, given the low frequency of HBV reactivation observed in a previous study (3.3%), the data of which should be taken with caution because it was obtained in a different context [54].

For HBsAg-positive patients, the choice between nucleo(t)side analogue prophylaxis and pre-emptive therapy is strongly in favor of the first option, although a long careful follow-up is then expected and is made more complex by the COVID-19 pandemic, which has led to severe restrictions on movements and economies, and it has given rise to the fear of being infected in healthcare facilities in a large part of the population.

In view of the low rate of HBV reactivation, pre-emptive therapy may be sufficient in principle for HBsAg-negative-HBcAb positive patients, provided that close HBsAg and HBV DNA monitoring is performed during immunosuppressive therapy and the long-term post-treatment follow-up.

The data analyzed in this review article lead us to identify corticosteroids and other immunosuppressive drugs administered to COVID-19 patients, rather than the actual COVID-19 infection per se, as the prevailing cause of HBV reactivation, an event that can and should be avoided by screening for HBV markers all COVID-19 patients and applying the appropriate preventive measures for both HBsAg-positive and HBsAg-negative/HBcAb-positive patients.

## Figures and Tables

**Figure 1 pathogens-11-00816-f001:**
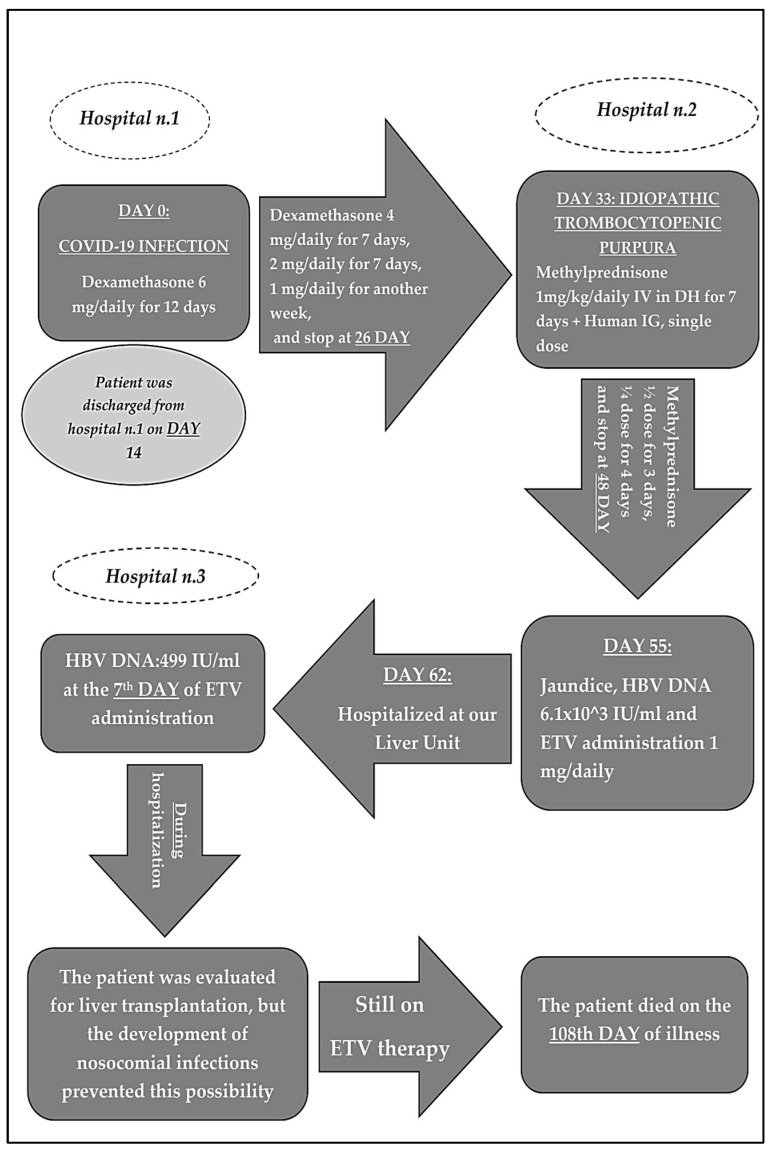
Clinical and therapeutical history of patient.

**Table 1 pathogens-11-00816-t001:** Summary of data on HBV reactivation in COVID-19 patients.

Author [Ref.]	Type of Study	Immunosuppressive Treatment for SARS-CoV-2 Infection	Sex	HBV Baseline Serology	HBV Treatment before Admission	Reactivation Timing from SARS-CoV-2 Infection	Age, Year	COVID-19 Severity	Outcome	HBV Therapy
Wu et al. [26]	Case Report	Methylprednisolone 40 mg /daily	M	HBsAg +, HBV DNA -	ETV	6 days	45	Low (fatigue, fever)	Return to normal liver Function	TDF Prophylaxis
Aldalheei et al. [27]	Case Report	None	M	Unavailable	None	Unavailable	36	Critical	Improved liver function	ETV prophylaxis (1mg/daily)
Rodriguez-Tajes et al. [28]	Prospective cohort of 38 patients	Various immunosuppressive drugs (mainly TCZ and corticosteroid)	70% M	HBsAg − / HBcAb +			69 median value	Prevalently Severe	No HBV reactivation	ETV
Rodriguez-Tajes et al. [28]	Prospective cohort of 23 patients	Various immunosuppressive drugs (mainly TCZ and corticosteroid)	74% M	HBsAg − / HBcAb +	None	30–60 days after last dose of im. sup. treatment	62 median value	Prevalently severe	8.7% Reactivation	No Prophylaxis

## Data Availability

The data presented in this study are available on request from the corresponding author.

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
