# Peer review of "COVID-19 as Another Trigger for HBV Reactivation: Clinical Case and Review of Literature"

_pathogens, 2022, doi:10.3390/pathogens11070816_

Round 1

Reviewer 1 Report

The manuscript by Sagnelli et al reports a fatal case of reactivated HBV infection in a patient following immunosuppressive treatment for COVID-19. It also provides a summary of previously published case reports of 2 patients with HBV infection associated with COVID-19 & a cohort of 61 patients with COVID-19 who had markers of HBV, 2 of whom reactivated HBV infection following immunosuppressive treatment for COVID-19. It is not clear if the 61 patients were HBV surface antibody (HBsAb)-positive & had cleared HBV infection prior to their COVID-19 infection or if they were HBV carriers with low level HBV infection. The manuscript would be strengthened by further details of this patient cohort.

It is clear that HBV reactivation can occur following immunosuppressive treatment for COVID-19 infection but the rate of reactivation is low. While this report of a single case provides new data that will complement the previous publications, it may not be of wide interest.

Comments, suggestions & proofreading edits are provided below.

1.       Line 2. Title. Consider revising to: “Immunosuppressive treatment of COVID-19 as another trigger for hepatitis B virus reactivation: clinical case …”

2.       Line 12. Provide full name for HBV.

3.       Line 14. Provide full name for HBsAg.

4.       Lines 37-40. How do the levels & duration of corticosteroid treatments reported in references [1-9] & [10-13] compare to those used in COVID-19 patients? Please comment on the impact this may have on HBV reactivation in the setting of COVID-19.

5.       Line 39. Provide full name for HBsAg.

6.       Line 40. Provide full description of anti-HBc.

7.       Line 45. Provide abbreviation for tocilizumab (TCZ).

8.       Lines 58-59. The statement that: “people with pre-existing HBV infection are at an increased risk of acquiring SARS-CoV-2 infection” seems difficult to prove given the high rates of SAS-CoV-2 infection in the general population. It also seems unlikely that HBV could affect transmission given the respiratory nature of SARS-CoV-2 infection. Is there any published or anecdotal evidence for this or any suggestion that HBV interacts with the SARS-CoV-2 receptor ACE2?

9.       Line 60. Was the HBV infection pre-existing?

10.   Line 61. Consider revising to: “in 14 patients their pre-existing liver injury worsened”.

11.   Line 63. Was the liver injury pre-existing?

12.   Line 74. Insert full stop after “occult”.

13.   Lines 76, 80, 84, 102, 121, 153, 159, 162, 184, 212. Delete hyphen from HBV-DNA.

14.   Lines 78-79. Please revise to clarify your meaning.

15.   Lines 80-112. Consider including a Figure that clearly sets out the timeline of COVID-19 infection & recovery, immunosuppressive therapy, nucleos(t)ide analogue therapy & death.

16.   Line 85. Revise to “never been treated with nucleos(t)ide analogues.”

17.   Line 86. What dose of dexamethasone?

18.   Line 87. What date was the patient discharged?

19.   Line 95. Replace “entecavir” with “ETV”. What date was ETV treatment started & how long did it continue? Replace “bid” with “twice/day” for consistency with Table 1.

20.   Line 98. Was the patient still receiving ETV on admission on August 4, 2021?

21.   Line 98-102. Provide the full name for ALT, AST, INR, PT, aPTT, d-dimer, HBsAb, HCV, HBeAg, HBeAb, HDV & HIV.

22.   Line 103. Replace “entecavir” with “ETV”. What were the dates at day 7 & day 15 of ETV therapy? Was the patient continued on ETV until the time of death? Replace “negative” with “not detected”.

23.   Line 103. Superscript 10^3 in “6.1 x 10^3 IU/ml”.

24.   Line 116. Delete heading: “4.1. Table 1”.

25.   Table 1. Correct the spelling of COVID-19 in the top row. Check the table for various spacing issues & lines above “Age, year”.

26.   Table 1 & line 151. Correct spelling of “prospective”. Was it a “prospective” or “retrospective” study? On line 151 It is referred to as a “perspective cohort study”.

27.   Table 1. Wu et al is reference [26] not [23].

28.   Table 1. Can the data from Rodriguez-Tajes et al [28] be divided into groups with & without nucleos(t)ide analogue therapy & with & without HBsAb? The description of sex & age is confusing.

29.   Line 118. Revise “chronic hepatitis B” to “chronic HBV infection”.

30.   Lines 118-128. Provide more details of the timing of hospitalization & COVID-19 recovery etc. Was the patient treated with ETV at the time of hospitalisation?

31.   Line 123. Replace “QD” with “daily” for consistency with Table 1.

32.   Line 124. Replace “serum aminotransferase values” with ALT, AST etc.

33.   Line 125. Superscript 10^2 in “1.11 x 10^2 IU/ml”.

34.   Line 139. Provide the full name for HAV, HEV.

35.   Line 141, 157. Insert brackets around dose of ETV & vitamin K.

36.   Line 149. Revise “acute hepatitis B” to “acute HBV infection”.

37.   Lines 152-153. Were the 600 HBV surface antigen (HBsAg)-negative, HBV core antibody (HBcAb)-positive & HBV DNA-negative patients tested for anti-HBs antibodies to determine if they had recovered from HBV infection & had reactivated their infection from residual HBV cccDNA or were HBV carriers without detectable levels of serum HBsAg or HBV DNA?

38.   Line 170, 192, 215. Check spelling of “COVID-19”.

39.   Line 242. Reference 2. Insert volume & page numbers.

40.   Line 260. Reference 9. Bold “2021”.

41.   Line 282, 327. Volume number should be fully italicised.

42.   References. Proof reading is required. Check spacing after year of publication, page numbers etc.  

Author Response

To the Editor in Chief of Pathogens

We re-submit our article “COVID-19 as another trigger for HBV reactivation: clinical case and review of literature, Special issue: Issue "Viral Hepatitis: The New Challenge in the Era of Antiviral Treatments".

The following changes (shown underlined). The manuscript has been improved according to the suggestions of the reviewer:

Reviewer(s)' Comments to Author:

Reviewer #1: The manuscript by Sagnelli et al reports a fatal case of reactivated HBV infection in a patient following immunosuppressive treatment for COVID-19. It also provides a summary of previously published case reports of 2 patients with HBV infection associated with COVID-19 & a cohort of 61 patients with COVID-19 who had markers of HBV, 2 of whom reactivated HBV infection following immunosuppressive treatment for COVID-19. It is not clear if the 61 patients were HBV surface antibody (HBsAb)-positive & had cleared HBV infection prior to their COVID-19 infection or if they were HBV carriers with low level HBV infection. The manuscript would be strengthened by further details of this patient cohort.

It is clear that HBV reactivation can occur following immunosuppressive treatment for COVID-19 infection but the rate of reactivation is low. While this report of a single case provides new data that will complement the previous publications, it may not be of wide interest.

Comments, suggestions & proofreading edits are provided below.

Point 2:  Line 2. Title. Consider revising to: “Immunosuppressive treatment of COVID-19 as another trigger for hepatitis B virus reactivation: clinical case …”

Answer to the Reviewer point 1: The observation of the reviewer has been accepted and the new manuscript has been  modified accordingly.

Point 2:  Line 12. Provide full name for HBV.

Answer to the Reviewer point 2: The observation of the reviewer has been accepted and the new manuscript has been  modified accordingly.

Point 3:     Line 14. Provide full name for HBsAg.

Answer to the Reviewer point 3: The observation of the reviewer has been accepted and the new manuscript has been  modified accordingly.

Point 4:     Lines 37-40. How do the levels & duration of corticosteroid treatments reported in references [1-9] & [10-13] compare to those used in COVID-19 patients? Please comment on the impact this may have on HBV reactivation in the setting of COVID-19.

Answer to the Reviewer point 4: The observation of the reviewer has been accepted and the new manuscript has been  modified accordingly.

Point 5:      Line 39. Provide full name for HBsAg.

Answer to the Reviewer point 5: The observation of the reviewer has been accepted and the new manuscript has been  modified accordingly.

Point 6: Line 40. Provide full description of anti-HBc.

Answer to the Reviewer point 6: The observation of the reviewer has been accepted and the new manuscript has been  modified accordingly.

Point 7:  Line 45. Provide abbreviation for tocilizumab (TCZ).

Answer to the Reviewer point 7: The observation of the reviewer has been accepted and the new manuscript has been  modified accordingly.

Point 8:  Lines 58-59. The statement that: “people with pre-existing HBV infection are at an increased risk of acquiring SARS-CoV-2 infection” seems difficult to prove given the high rates of SAS-CoV-2 infection in the general population. It also seems unlikely that HBV could affect transmission given the respiratory nature of SARS-CoV-2 infection. Is there any published or anecdotal evidence for this or any suggestion that HBV interacts with the SARS-CoV-2 receptor ACE2?

Answer to the Reviewer point 8: The observation of the reviewer has been accepted and the new manuscript has been  modified accordingly.

Point 9:    Line 60. Was the HBV infection pre-existing?

 Answer to the Reviewer point 9: The observation of the reviewer has been accepted and the new manuscript has been  modified accordingly.

 Point 10:     Line 61. Consider revising to: “in 14 patients their pre-existing liver injury worsened”.

Answer to the Reviewer point 10: The observation of the reviewer has been accepted and the new manuscript has been  modified accordingly.

Point 11:   Line 63. Was the liver injury pre-existing?

Answer to the Reviewer point 1: The observation of the reviewer has been accepted and the new manuscript has been  modified accordingly.

Point 12:    Line 74. Insert full stop after “occult”.

Answer to the Reviewer point 12 The observation of the reviewer has been accepted and the new manuscript has been  modified accordingly.

Point 13:    Lines 76, 80, 84, 102, 121, 153, 159, 162, 184, 212. Delete hyphen from HBV-DNA.

Answer to the Reviewer point 13: The observation of the reviewer has been accepted and the new manuscript has been  modified accordingly.

Point 14:   Lines 78-79. Please revise to clarify your meaning.

Answer to the Reviewer point 14: The observation of the reviewer has been accepted and the new manuscript has been  modified accordingly.

Point 15:   Lines 80-112. Consider including a Figure that clearly sets out the timeline of COVID-19 infection & recovery, immunosuppressive therapy, nucleos(t)ide analogue therapy & death.

Answer to the Reviewer point 15: The observation of the reviewer has been accepted and a Figure 1 was add in the new manuscript has been  modified accordingly.

Point 16:   Line 85. Revise to “never been treated with nucleos(t)ide analogues.”

Answer to the Reviewer point 16: The observation of the reviewer has been accepted and the new manuscript has been  modified accordingly.

Point 17:  Line 86. What dose of dexamethasone?

 Answer to the Reviewer point 17: The observation of the reviewer has been accepted and the new manuscript has been  modified accordingly.

Point 18:   Line 87. What date was the patient discharged?

Answer to the Reviewer point 18: The observation of the reviewer has been accepted and the new manuscript has been  modified accordingly, we add Figure 1.

Point 19:     Line 95. Replace “entecavir” with “ETV”. What date was ETV treatment started & how long did it continue? Replace “bid” with “twice/day” for consistency with Table 1.

Answer to the Reviewer point 19: The observation of the reviewer has been accepted and the new manuscript has been  modified accordingly. data added to the text as requested.

Point 20:   Line 98. Was the patient still receiving ETV on admission on August 4, 2021?

Answer to the Reviewer point 20: The observation of the reviewer has been accepted and the new manuscript has been  modified accordingly.

Point 21:   Line 98-102. Provide the full name for ALT, AST, INR, PT, aPTT, d-dimer, HBsAb, HCV, HBeAg, HBeAb, HDV & HIV.

Answer to the Reviewer point 21: The observation of the reviewer has been accepted and the new manuscript has been  modified accordingly.

Point 22:     Line 103. Replace “entecavir” with “ETV”. What were the dates at day 7 & day 15 of ETV therapy? Was the patient continued on ETV until the time of death? Replace “negative” with “not detected”.

Answer to the Reviewer point 22: The observation of the reviewer has been accepted and the new manuscript has been  modified accordingly.

Point 23:     Line 103. Superscript 10^3 in “6.1 x 10^3 IU/ml”.

Answer to the Reviewer point 23: The observation of the reviewer has been accepted and the new manuscript has been  modified accordingly.

Point 24:   Line 116. Delete heading: “4.1. Table 1”.

Answer to the Reviewer point 24: The observation of the reviewer has been accepted and the new manuscript has been  modified accordingly.

Point 25:  Table 1. Correct the spelling of COVID-19 in the top row. Check the table for various spacing issues & lines above “Age, year”.

Answer to the Reviewer point 25: The observation of the reviewer has been accepted and the new manuscript has been  modified accordingly.

Point 26:    Table 1 & line 151. Correct spelling of “prospective”. Was it a “prospective” or “retrospective” study? On line 151 It is referred to as a “perspective cohort study”.

Answer to the Reviewer point 26: The observation of the reviewer has been accepted and the new manuscript has been  modified accordingly.

Point 27:    Table 1. Wu et al is reference [26] not [23].

Answer to the Reviewer point 27: The observation of the reviewer has been accepted and the new manuscript has been  modified accordingly.

Point 28:  Table 1. Can the data from Rodriguez-Tajes et al [28] be divided into groups with & without nucleos(t)ide analogue therapy & with & without HBsAb? The description of sex & age is confusing.

Answer to the Reviewer point 28: The observation of the reviewer has been accepted and the new manuscript has been  modified accordingly.

Point 29:  .   Line 118. Revise “chronic hepatitis B” to “chronic HBV infection”.

Answer to the Reviewer point 29: The observation of the reviewer has been accepted and the new manuscript has been  modified accordingly.

Point 30:   Lines 118-128. Provide more details of the timing of hospitalization & COVID-19 recovery etc. Was the patient treated with ETV at the time of hospitalisation?

Answer to the Reviewer point 30: The observation of the reviewer has been accepted and the new manuscript has been  modified accordingly.

Point 31:   Line 123. Replace “QD” with “daily” for consistency with Table 1.

Answer to the Reviewer point 31: The observation of the reviewer has been accepted and the new manuscript has been  modified accordingly.

Point 32:  Line 124. Replace “serum aminotransferase values” with ALT, AST etc.

Answer to the Reviewer point 32: The observation of the reviewer has been accepted and the new manuscript has been  modified accordingly.

Point 33:     Line 125. Superscript 10^2 in “1.11 x 10^2 IU/ml”.

Answer to the Reviewer point 33: The observation of the reviewer has been accepted and the new manuscript has been  modified accordingly.

Point 34:   Line 139. Provide the full name for HAV, HEV.

Answer to the Reviewer point 34: The observation of the reviewer has been accepted and the new manuscript has been  modified accordingly.

Point 35:   Line 141, 157. Insert brackets around dose of ETV & vitamin K.

Answer to the Reviewer point 35: The observation of the reviewer has been accepted and the new manuscript has been  modified accordingly.

Point 36:   Line 149. Revise “acute hepatitis B” to “acute HBV infection”.

Answer to the Reviewer point 36: The observation of the reviewer has been accepted and the new manuscript has been  modified accordingly.

Point 37:   Lines 152-153. Were the 600 HBV surface antigen (HBsAg)-negative, HBV core antibody (HBcAb)-positive & HBV DNA-negative patients tested for anti-HBs antibodies to determine if they had recovered from HBV infection & had reactivated their infection from residual HBV cccDNA or were HBV carriers without detectable levels of serum HBsAg or HBV DNA?

Answer to the Reviewer point 37: The observation of the reviewer has been accepted and the new manuscript has been  modified accordingly.

Point 38:   Line 170, 192, 215. Check spelling of “COVID-19”.

Answer to the Reviewer point 38: The observation of the reviewer has been accepted and the new manuscript has been  modified accordingly.

Point 39:    Line 242. Reference 2. Insert volume & page numbers.

Answer to the Reviewer point 39: Insert volume & page numbers. Are not available. this reference is available at on line site: Recent Advances in HBV Reactivation Research - PubMed (nih.gov)

Point 40:    Line 260. Reference 9. Bold “2021”.

Answer to the Reviewer point 40: The observation of the reviewer has been accepted and the new manuscript has been  modified accordingly.

Point 41:   Line 282, 327. Volume number should be fully italicised.

Answer to the Reviewer point 41: The observation of the reviewer has been accepted and the new manuscript has been  modified accordingly.

Point 42:    References. Proof reading is required. Check spacing after year of publication, page numbers etc.  

Answer to the Reviewer point 42: The observation of the reviewer has been accepted and the new manuscript has been  modified accordingly.

We thank the Editor and the Reviewers for helping us to improve our paper.

The manuscript has been read and approved by all the authors.

We also declare that we have no conflict of interest in connection with this paper.

We sincerely hope that the enclosed manuscript can be accepted for publication in the: Pathogens

Prof.ssa Caterina Sagnelli

Reviewer 2 Report

The paper focus on HBV management in COVID-19 disease. The topic is actual with important implication in the clinical practice. Although the paper is well written, there are some points that need to be addressed.

 INTRODUCTION AND REVIEW OF THE LITERATURE

Major points:

Table 1:

-your case report should not be included in the table

-cause of reactivation: change column title into “Immunosoppressive treatment for sars-cov2 infection” and add dosage and length of immunosuppressive treatment

-add columns for timing for reactivation, HBsAg/HBV DNA at baseline, ongoing treatment for HBV infection at baseline

-outcome should be focused on liver outcome

-column COVID severity: determine COVID-severity use NIH score or other validated ones like NEWS or qCSI

--Ref 23 (Wu et al) is ref 26, please correct

Paragraph 2: As you mention in paragraph 4, also here it should be clear in this paragraph that the risk for HBV reactivation in HBsAg neg/HBcAb pos is low in the setting of COVID19, pending additional studies.  

Paragraph 4: the paragraph is too long; the majority of information reported could be included in table 1 .

--lines 173-175 this argument should be developed, especially for steroid treatment that is very important in the setting of Sars-cov2. Particularly it could be interesting argument on duration and dosing for steroid treatment. Add also one comment on HBV DNA at baseline

--lines 180-187:  references?

Minor points

-lines 39-40: please specify that HBV reactivation is less frequent in HBsAg negative individuals

-lines 44: COVID-19 offers.. : I would not use the simple present; treatment with steroids is usually short-term with this indication and available evidence is still low

-lines 52-56: I think that the different endemicity of HBV according to country of origin could be better defined by other paper, even not focused in COVID-19 patients.

-line 118 : reference

-please review English language: particularly lines 57-59, 194-201

 -CASE REPORT:  

--did the patient sign an informed consent for data use for research scopes?

--“The patient had been chronically infected with HBV since 1986 but being HBV-DNA-negative with

normal 84 serum aminotransferases and he had never been treated” (HBsAg, HBcAb?). It would be

helpful to indicate HIV serology and laboratory data such as kidney function, liver function,

coagulation at baseline.

--“During hospitalization, he received dexamethasone”:pro kg dose?, de-escalation?, how long he took

Corticosteroid?

--“Thirty days after discharge”it would be better to put the time from steroid suspension

--alcohol use? Concomitant causes for liver injury?

--did he take anti-sars-cov2 antivirals that could give liver toxicity?

--another city hospital: I think it is unnecessary

--at the time of acute liver failure, was HBV DNA tested before entecavir administration?  Why 0.5 bid?

--June 28, August 4 ecc: please avoid dated and mention timing from sars-cov2 infection , because patient could be identified with these information

-which data corroborates the diagnosis of acute autoimmune thrombocytopenia?

Author Response

To the Editor in Chief of Pathogens

We re-submit our article “COVID-19 as another trigger for HBV reactivation: clinical case and review of literature, Special issue: Issue "Viral Hepatitis: The New Challenge in the Era of Antiviral Treatments".

The following changes (shown underlined). The manuscript has been improved according to the suggestions of the reviewer:

Reviewer(s)' Comments to Author:

Reviewer #2: The paper focus on HBV management in COVID-19 disease. The topic is actual with important implication in the clinical practice. Although the paper is well written, there are some points that need to be addressed.

Point 1:   INTRODUCTION AND REVIEW OF THE LITERATURE

Major points:

Table 1:

- your case report should not be included in the table

- cause of reactivation: change column title into “Immunosoppressive treatment for sars-cov2 infection” and add dosage and length of immunosuppressive treatment

- add columns for timing for reactivation, HBsAg/HBV DNA at baseline, ongoing treatment for HBV infection at baseline

- outcome should be focused on liver outcome

-column COVID severity: determine COVID-severity use NIH score or other validated ones like NEWS or qCSI -Ref 23 (Wu et al) is ref 26, please correct

-Paragraph 2: As you mention in paragraph 4, also here it should be clear in this paragraph that the risk for HBV reactivation in HBsAg neg/HBcAb pos is low in the setting of COVID19, pending additional studies.  

-  Paragraph 4: the paragraph is too long; the majority of information reported could be included in table 1 . Caterina

--lines 173-175 this argument should be developed, especially for steroid treatment that is very important in the setting of Sars-cov2. Particularly it could be interesting argument on duration and dosing for steroid

treatment. Add also one comment on HBV DNA at baseline

--lines 180-187:  references

Answer to the Reviewer point 1: The observation of the reviewer has been accepted and the new manuscript has been  modified accordingly.

Point 2: Minor points

-lines 39-40: please specify that HBV reactivation is less frequent in HBsAg negative individuals

Answer to the Reviewer point 3: The observation of the reviewer has been accepted and the new manuscript has been  modified accordingly.

Point 3:  -lines 44: COVID-19 offers.. : I would not use the simple present; treatment with steroids is usually short-term with this indication and available evidence is still low

Answer to the Reviewer point 3: The observation of the reviewer has been accepted and the new manuscript has been  modified accordingly.

Point 4:  -lines 52-56: I think that the different endemicity of HBV according to country of origin could be better defined by other paper, even not focused in COVID-19 patients.

Answer to the Reviewer point 4: The observation of the reviewer has been accepted and the new manuscript has been  modified accordingly.

Point 5:  -line 118 : reference

Answer to the Reviewer point 5: The observation of the reviewer has been accepted and the new manuscript has been  modified accordingly.

Point 6:  -please review English language: particularly lines 57-59, 194-201

Answer to the Reviewer point 6: The observation of the reviewer has been accepted and the new manuscript has been  modified accordingly.

Point 7:   -CASE REPORT:  

--did the patient sign an informed consent for data use for research scopes?

--“The patient had been chronically infected with HBV since 1986 but being HBV-DNA-negative with

normal 84 serum aminotransferases and he had never been treated” (HBsAg, HBcAb?). It would be

helpful to indicate HIV serology and laboratory data such as kidney function, liver function, coagulation at baseline.

--“During hospitalization, he received dexamethasone”:pro kg dose?, de-escalation?, how long he took

Corticosteroid?

--Thirty days after dischargeӈit would be better to put the time from steroid suspension

--alcohol use? Concomitant causes for liver injury?

--did he take anti-sars-cov2 antivirals that could give liver toxicity?

--another city hospital: I think it is unnecessary

--at the time of acute liver failure, was HBV DNA tested before entecavir administration?  Why 0.5 bid?

--June 28, August 4 ecc: please avoid dated and mention timing from sars-cov2 infection , because patient could be identified with these information:

-which data corroborates the diagnosis of acute autoimmune thrombocytopenia?

Answer to the Reviewer point 7: The observation of the reviewer has been accepted and the new manuscript has been  modified accordingly.

We thank the Editor and the Reviewers for helping us to improve our paper.

The manuscript has been read and approved by all the authors.

We also declare that we have no conflict of interest in connection with this paper.

We sincerely hope that the enclosed manuscript can be accepted for publication in the: Pathogens

Prof.ssa Caterina Sagnelli

Round 2

Reviewer 1 Report

The revised manuscript has been significantly improved but still requires careful proof reading and Figure 1 needs to be extensively revised.

Line 41. Close gap in “HBsAg-positive”.

Line 42, Revise “anti-HBc” to “HBcAb” to match later sections.

Line 43. Delete “of”.

Line 45. Do you mean “exceeded” rather than “overcome”?

Line 46. Check do you really mean refs [1-13]?

Line 60. Revise to “or whether”.

Line 80, 84, 197. Delete hyphen from HBV-DNA.

Line 87. Patient was hospitalized on “day 0” but this is shown as “day 1” in Figure 1.

Line 90. Revise to “patient”.

Line 91-94. Revise to use lower case for hepatitis, and in all cases hyphenate -positive and -negative.

Line 94. Remove “the” in front of “Human”.

Line 95 and 123. Provide full names for AST, ALT and use abbreviations later in the manuscript.

Line 98. Delete “of” and revise 6 mg daily to 6 mg/day.

Line 101-102. Revise “daisies” to “days” ?, insert space in “4 mg” and “1 mg”, replace “week” with intervals in days.

Line 104. Define “ARDS”.

Figure 1. Top left box. Define this as hospital #1. Correct the spelling of “Dexamethasone 6 mg/day”.

Figure 1. Top right box. Indicate that the patient was discharged from Hospital #1 on day 14. Define “Stop cortisone descaling” in 2 positions. This is not mentioned in the text. Under “Day 23” define “IPT”. Include details of dexamethasone dosing and timing.

Figure 1. Bottom box. Maybe break this box into 2 parts, to define Hospitals #2 and #3. Include HBV DNA titers in balloons for Day 55, Day 62, and Day 70. In Day 55 balloon revise to “1 mg/day”. In Day 62 balloon check spelling of “our Liver Unit”. Is this Hospital #3?

Line 119. Revise to “severe”.

Line 126-128. Revise to “Serological tests indicated the patient was HBsAg-positive, HBsAb-negative etc” taking care to hyphenate each -positive and -negative serological marker.

Line 135. Revise to “blood chemistry was performed.”

Table 1. Revise “40 mg q24hr”. Remove stray line above “COVID-19 Severity”. Check “ETV Prophylaxis (1 mg/day)”. Define “(CS)”. Add “HBsAg – HBcAb + to group of 38 patients.

Line 161-162. Hyphenate “HBeAg-, HBeAb- and HBV DNA-negative”.

Line 173-174.Revise to: “The patient was HBsAg-, HBcAb IgM- and HBeAb-negative”.

Line 195-196. Delete “In detail” and “i in detail”.

Line 209-219 and Lines 235-245 are a repeat.

Line 258. Use all caps for COVID-19.

Author Response

To the Editor in Chief of Pathogens

We re-submit our article “COVID-19 as another trigger for HBV reactivation: clinical case and review of literature, Special issue: Issue "Viral Hepatitis: The New Challenge in the Era of Antiviral Treatments".

The following changes (shown underlined). The manuscript has been improved according to the suggestions of the reviewer:

Reviewer(s)' Comments to Author:
Reviewer #1: The revised manuscript has been significantly improved but still requires careful proof reading and Figure 1 needs to be extensively revised.

Point 1:

Line 41. Close gap in “HBsAg-positive”.

Line 42, Revise “anti-HBc” to “HBcAb” to match later sections.

Line 43. Delete “of”.

Line 45. Do you mean “exceeded” rather than “overcome”?

Line 46. Check do you really mean refs [1-13]?

Line 60. Revise to “or whether”.

Line 80, 84, 197. Delete hyphen from HBV-DNA.

Line 87. Patient was hospitalized on “day 0” but this is shown as “day 1” in Figure 1.

Line 90. Revise to “patient”.

Line 91-94. Revise to use lower case for hepatitis, and in all cases hyphenate -positive and -negative.

Line 94. Remove “the” in front of “Human”.

Line 95 and 123. Provide full names for AST, ALT and use abbreviations later in the manuscript.

Line 98. Delete “of” and revise 6 mg daily to 6 mg/day.

Line 101-102. Revise “daisies” to “days” ?, insert space in “4 mg” and “1 mg”, replace “week” with intervals in days.

Line 104. Define “ARDS”.

Figure 1. Top left box. Define this as hospital #1. Correct the spelling of “Dexamethasone 6 mg/day”.

Figure 1. Top right box. Indicate that the patient was discharged from Hospital #1 on day 14. Define “Stop cortisone descaling” in 2 positions. This is not mentioned in the text.

Under “Day 23” define “IPT”. Include details of dexamethasone dosing and timing.

Figure 1. Bottom box. Maybe break this box into 2 parts, to define Hospitals #2 and #3. Include HBV DNA titers in balloons for Day 55, Day 62, and Day 70. In Day 55 balloon revise to “1 mg/day”. In Day 62 balloon check spelling of “our Liver Unit”. Is this Hospital #3?

Line 119. Revise to “severe”.

Line 126-128. Revise to “Serological tests indicated the patient was HBsAg-positive, HBsAb-negative etc” taking care to hyphenate each -positive and -negative serological marker.

Line 135. Revise to “blood chemistry was performed.”

Table 1. Revise “40 mg q24hr”. Remove stray line above “COVID-19 Severity”. Check “ETV Prophylaxis (1 mg/day)”. Define “(CS)”. Add “HBsAg – HBcAb + to group of 38 patients.

Line 161-162. Hyphenate “HBeAg-, HBeAb- and HBV DNA-negative”.

Line 173-174.Revise to: “The patient was HBsAg-, HBcAb IgM- and HBeAb-negative”.

Line 195-196. Delete “In detail” and “i in detail”.

Line 209-219 and Lines 235-245 are a repeat.

Line 258. Use all caps for COVID-19.

Answer to the Reviewer point 1: The observation of the reviewer has been accepted and the new manuscript has been  modified accordingly.

We thank the Editor and the Reviewers for helping us to improve our paper.

The manuscript has been read and approved by all the authors.

We also declare that we have no conflict of interest in connection with this paper.

We sincerely hope that the enclosed manuscript can be accepted for publication in the: Pathogens

Prof.ssa Caterina Sagnelli

Reviewer 2 Report

The paper is well written and all observations have ben addressed.

There are still some typo errors (i.e. Lines 100-105; 195-196)

Table 1: reactivation time: from sars cov2infection or from starting oh corti o steroidi? Please precise it.

No other comments.

Author Response

To the Editor in Chief of Pathogens

We re-submit our article “COVID-19 as another trigger for HBV reactivation: clinical case and review of literature, Special issue: Issue "Viral Hepatitis: The New Challenge in the Era of Antiviral Treatments".

The following changes (shown underlined). The manuscript has been improved according to the suggestions of the reviewer:

Reviewer(s)' Comments to Author:

Reviewer #2: The paper is well written and all observations have ben addressed.

Point 1:  There are still some typo errors (i.e. Lines 100-105; 195-196)

Answer to the Reviewer point 1: The observation of the reviewer has been accepted and the new manuscript has been  modified accordingly.

Point 2:  Table 1: reactivation time: from sars cov2infection or from starting oh corti o steroidi? Please precise it.

No other comments.

Answer to the Reviewer point 2: The observation of the reviewer has been accepted and the new manuscript has been  modified accordingly.

We thank the Editor and the Reviewers for helping us to improve our paper.

The manuscript has been read and approved by all the authors.

We also declare that we have no conflict of interest in connection with this paper.

We sincerely hope that the enclosed manuscript can be accepted for publication in the: Pathogens

Prof.ssa Caterina Sagnelli
